# An Update on the Chemokine System in the Development of NAFLD

**DOI:** 10.3390/medicina58060761

**Published:** 2022-06-05

**Authors:** Naoto Nagata, Guanliang Chen, Liang Xu, Hitoshi Ando

**Affiliations:** 1Department of Cellular and Molecular Function Analysis, Graduate School of Medical Sciences, Kanazawa University, Kanazawa 920-8640, Japan; h-ando@med.kanazawa-u.ac.jp; 2Department of Obstetrics and Gynecology, Keio University School of Medicine, 35 Shinanomachi, Shinjuku-ku, Tokyo 160-8582, Japan; guanliangc@keio.jp; 3School of Laboratory Medicine and Life Sciences, Wenzhou Medical University, Wenzhou 325035, China; liangxu1023@gmail.com

**Keywords:** chemokine, nonalcoholic fatty liver disease, nonalcoholic steatohepatitis, inflammation, immune cells

## Abstract

Nonalcoholic fatty liver disease (NAFLD) is the most common chronic liver disease in the world. Sustained hepatic inflammation is a key driver of the transition from simple fatty liver to nonalcoholic steatohepatitis (NASH), the more aggressive form of NAFLD. Hepatic inflammation is orchestrated by chemokines, a family of chemoattractant cytokines that are produced by hepatocytes, Kupffer cells (liver resident macrophages), hepatic stellate cells, endothelial cells, and vascular smooth muscle cells. Over the last three decades, accumulating evidence from both clinical and experimental investigations demonstrated that chemokines and their receptors are increased in the livers of NAFLD patients and that CC chemokine ligand (CCL) 2 and CCL5 in particular play a pivotal role in inducing insulin resistance, steatosis, inflammation, and fibrosis in liver disease. Cenicriviroc (CVC), a dual antagonist of these chemokines’ receptors, CCR2 and CCR5, has been tested in clinical trials in patients with NASH-associated liver fibrosis. Additionally, recent studies revealed that other chemokines, such as CCL3, CCL25, CX3C chemokine ligand 1 (CX3CL1), CXC chemokine ligand 1 (CXCL1), and CXCL16, can also contribute to the pathogenesis of NAFLD. Here, we review recent updates on the roles of chemokines in the development of NAFLD and their blockade as a potential therapeutic approach.

## 1. Introduction

Chemokines (Greek—kinos, movement) are a large family of chemotactic cytokines that involve immune and inflammatory responses through the chemoattraction and activation of leukocytes [1]. These small proteins (approximately 8–12 kilodaltons) are classified into four different subfamilies (CC, CXC, CX3C and XC) based on the presence of four cysteine residues in the conserved locations of *N*-terminals that are key to forming their 3-dimensional shape [2]. To date, approximately 50 chemokines expressed in various cell types and tissues have been identified in humans and mice [3]. In the liver, not only Kupffer cells but also hepatocytes, hepatic stellate cells (HSCs), liver sinusoidal endothelial cells, and vascular smooth muscle cells can secrete chemokines upon activation [4].

Chemokine receptors are a group of ~20 typical G protein-coupled seven-transmembrane proteins and are expressed in various leucocytes and immune cells. The directed migration of specific chemokine receptor-expressing cells allows for their recruitment along a chemokine concentration gradient [5]. Upon ligand binding, chemokine receptors mediate cellular calcium influx through phosphatidylinositol 3-kinase and small Rho guanosine triphosphatase activation, thereby increasing the avidity of leukocyte integrins that promote leukocytes’ interactions with intercellular adhesion molecules on sinusoidal endothelial cells [6,7]. Chemokines regulate not only immune cell recruitment during inflammation through inflammatory chemokines (CCL2, CCL3, CCL5, etc.) but also the trafficking of innate immune cells at homeostasis through homeostatic chemokines (CXCL12, etc.); they also modulate the functions of nonimmune cells, such as fibrogenic HSCs [8,9,10].

Regarding the pathogenesis of NAFLD, relevant chemokines and their receptors have been well summarized by excellent reviews [11,12]. Particularly, the pathophysiological roles of CCL2 and CCL5 in the development of NAFLD have been well studied in both NAFLD patients and animal models. In NAFLD patients, elevated serum and hepatic mRNA levels of CCL2 increase the recruitment of CCR2-positive bone marrow-derived monocytes into the liver, resulting in further hepatic inflammation, fibrosis, and steatosis [13,14,15]. Accordingly, the genetic deletion or pharmacological inhibition of CCR2 has been reported to improve NASH and insulin resistance in mice [16,17]. CCL5 production is also increased by excessive lipid accumulation in the liver [18]. CCL5 is required for the progression of liver fibrosis by binding to CCR1 on liver macrophages and CCR5 on hepatic stellate cells [19,20]. Based on these observations, cenicriviroc (CVC), a dual CCR2 and CCR5 antagonist, is expected to improve NASH and has been tested in clinical trials in patients with NASH-associated liver fibrosis [21,22].

In this review, we will highlight recent updates on the roles of chemokines, including CCL3, CCL25, CXCL1, CXCL16, and CX3CL1 (Figure 1), in the development of NAFLD and their blockade as a potential therapeutic approach.

## 2. An Update on the Chemokine System in the Development of NAFLD

### 2.1. CCL3-CCR1 and CCR5

CCL3 (known as macrophage inflammatory protein-1α) is expressed in macrophages and secreted to recruit macrophages themselves, various leukocyte subtypes, and T cells to inflamed sites [23,24]. Various proinflammatory stimuli, such as viral infections, lipopolysaccharide, tumor necrosis factor-α (TNF-α), interferon-γ, and interleukin-1β (IL-1β), can induce the expression of CCL3 [25,26]. CCL3 signals through its receptors CCR1 and CCR5. T cells, eosinophils, and neutrophils express CCR1 [27]. NK cells and subsets of resting memory T cells, including some but not all Th1 cells, express CCR5 [27]. Monocytes and mature macrophages express both CCR1 and CCR5 [28].

The progression of some inflammatory diseases, including atherosclerosis [29], rheumatoid arthritis [30], and NAFLD [31,32], is associated with the increased expression of CCL3 and its receptors. Circulating CCL3 levels were significantly higher in NASH patients compared with healthy controls [31,32]. Recently, we also reported that the serum and hepatic protein levels of CCL3 were significantly higher in patients with NASH compared with those of healthy controls [33]. Unlike in a previous report [31,32], however, patients with simple fatty liver also showed a trend of increased serum CCL3 levels and a significant increase in hepatic CCL3 protein expression compared with those of healthy controls [33]. Additionally, the circulating levels of CCL3 were high according to the histological severity of ballooning, steatosis, and lobular inflammation [33]. These results suggest that CCL3 might be a causative factor, not just a result of advanced fibrosis, in NAFLD development.

We further investigated the role of CCL3 in the pathogenesis of NAFLD in mice fed a high-cholesterol and high-fat (CL) diet, a dietary model of NASH [34]. We found that the circulating levels and hepatic expression of CCL3 were elevated in the CL diet-fed mice and that the hepatic source of CCL3 was particularly M1-like macrophages rather than M2-like macrophages and other cell types [33]. The genetic deletion of CCL3 attenuated the CL diet-induced steatohepatitis and hepatic insulin resistance, at least partly, by decreasing macrophage recruitment and restoring alternative macrophage activation in the liver [33]. Moreover, the specific deletion of CCL3 in bone marrow cells eased CL diet-induced steatohepatitis [33]. These results suggest that CCL3 plays a certain role in the recruitment of bone marrow-derived monocytes into the liver and the M1 polarization of liver macrophages, which contributes to chronic inflammation and hepatic insulin resistance in the development of NAFLD.

### 2.2. CCL25-CCR9

The chemokine CCL25 is selectively and constitutively expressed in the thymus and small intestine. CCR9, the sole functional receptor of CCL25 [35], is expressed on thymocytes and intestinal lymphocytes [36]. The CCL25-CCR9 axis is crucial for mucosal lymphocyte recruitment to the small intestine followed by accumulating CCR9^+^CD4^+^ tissue-infiltrating T cells in both Crohn’s disease and a murine model of inflammatory bowel disease [37,38,39]. With regard to liver immunology, CCR9^+^ macrophages play a pathogenic role in a murine acute hepatitis model and humans [40]. Peripheral blood samples from patients with acute hepatitis had more TNF-α-producing CCR9^+^ monocytes than healthy volunteers [40]. Similarly, in concanavalin A-injected mice, bone marrow-derived CCR9^+^ macrophages accumulate in the liver, which produces high levels of TNF-α and promotes the Th1 differentiation of naive CD4^+^ T cells, thereby contributing to acute liver inflammation [40]. Additionally, Morikawa et al. provided multiple lines of evidence indicating that the CCL25-CCR9 axis also plays a pivotal role in NASH pathogenesis [41]: (1)Serum CCL25 and hepatic CCR9 and CCL25 levels were higher in patients with NASH compared with healthy volunteers and patients with simple fatty liver. (2) CCL25 was expressed in CD31^+^/LYVE1^+^ sinusoidal endothelial cells, whereas CCR9 was expressed in CD68^+^ macrophages and GFAP^+^/α-SMA^+^ HSCs in the livers of patients with NASH, and the numbers of these CCR9^+^ cells were significantly lower in the control samples. (3) CCR9-deficient mice showed alleviated diet-induced steatohepatitis associated with the decrease in the amount of CD11b^+^ inflammatory macrophage accumulation in the liver. (4) Consistent with human NASH, CCR9 was also expressed on HSCs in NASH mice and CCR9-deficient HSCs show fewer fibrogenic phenotypes. Finally (5) A CCR9 antagonist, vercirnon (CCX282-B) ameliorated steatohepatitis and the development of diethylnitrosamine-induced hepatocellular carcinoma in a high-fat diet-fed mice. These results indicate a therapeutic potential of CCR9 blockade in NAFLD.

### 2.3. CXCL1-CXCR2

CXCL1 is one of the major chemoattractants for neutrophils [42]. After binding to its receptor CXCR2, CXCL1 activates PI3K/Akt, MAP kinases, or phospholipase-β signaling pathways, increasing the recruitment of neutrophils into inflamed sites [43]. CXCL1 is also involved in the processes of wound healing, angiogenesis, tumorigenesis, and cell motility [44]. CXCL1 is highly expressed in the liver of NASH patients but not in the simple fatty livers in obese individuals or in high-fat diet (HFD)-fed mice [45,46]. In the choline-deficient amino acid-defined (CDAA) diet-induced mouse NASH model, the hepatic mRNA levels of CXCL1 are increased in a toll-like receptor 4 (TLR4)-MyD88-dependent manner, resulting in increased neutrophil infiltration associated with hepatic inflammation and fibrosis [47]. Additionally, adenoviral overexpression of CXCL1 in the liver is sufficient to activate progression from steatosis to steatohepatitis in HFD-fed mice by inducing hepatic neutrophile infiltration, oxidative stress, and hepatocyte apoptosis [48]. These studies indicate the importance of CXCL1-/CXCR2-mediated neutrophile recruitment during NAFLD development.

### 2.4. CXCL16-CXCR6

In conjunction with CD4, CXCR6 can serve as a co-receptor for the entry of human and most simian immunodeficiency viruses (human immunodeficiency virus type I and simian immunodeficiency virus) [49]. Similar to CCR5 and CXCR3, the expression pattern of CXCR6 is restricted to memory/effector T cells such as natural killer T (NKT) cells [50,51] and CD8^+^ T cells [52]. In the liver, CXCR6^+^ NKT cells patrol liver sinusoids and provide the intravascular immune surveillance of pathogens [53]. CXCL16, a membrane-bound ligand for CXCR6, is expressed on the hepatocytes and biliary epithelial cells in the portal tracts and on sinusoidal cells in both normal and chronically inflamed liver tissue such as hepatitis C [54]. CXCL16 promotes the adhesion of CXCR6^+^ cells to cholangiocytes and hepatocytes by triggering the conformational activation of β1 integrins and the binding to vascular cell adhesion molecule-1 (VCAM-1), thereby promoting liver inflammation [54]. Regarding NAFLD, Jing et al. demonstrated that serum levels of CXCL16 were elevated in NAFLD patients and that CXCL16 was strongly expressed around the steatotic hepatocytes in liver biopsy specimens [55]. Additionally, in the co-culture of murine hepatocytes and HSCs, lentiviral overexpression of CXCL16 increased lipid accumulation and mitochondrial stress in hepatocytes and induced the activation and proliferation of HSCs [55], suggesting that the CXCL16-CXCR6 axis mediates the crosstalk between hepatocytes and HSCs in NAFLD development.

In the liver, CXCR6 is also expressed in CD8^+^ T cells. The auto-aggression of CD8^+^ T cells may be involved in the development of hepatocellular carcinoma from NASH. CXCR6^+^ CD8^+^ T cells accumulate in the livers of a preclinical mouse model of NASH (mice fed a choline-deficient and HFD) and of patients with NASH [52]. The T cells are susceptible to metabolic stimuli such as acetate and extracellular ATP, showing auto-aggressive killing of cells in an MHC-class-I-independent fashion [52].

### 2.5. CX3CL1-CX3CR1

CX3CL1, also known as fractalkine, a membrane-anchored chemokine, is expressed on epithelial cells, dendritic cells, and neurons and could be induced by inflammatory cytokines, such as TNF-α and IFN-γ [56,57,58,59,60,61]. CX3CL1 drives integrin-dependent adhesion and promotes the retention of specific CX3CR1-expressing leukocytes. The receptor is mainly expressed on circulating monocytes, tissue-resident macrophages, dendritic cells, and T cells [56,62,63]. The *N*-terminal domain of CX3CL1, containing a CX3C motif, can be cleaved by ADAM Metallopeptidase Domain 10 (ADAM10) [64] and ADAM17 [65], yielding a soluble form that also ligates CX3CR1 and exerts potent chemotactic activity [66].

Similar to the other chemokines, both animal and clinical studies have demonstrated that CX3CL1-CX3CR1 signaling is enhanced in various inflammatory diseases, such as rheumatoid arthritis [67], atherosclerosis [68,69], and chronic hepatitis C [70]. However, the pathophysiological role of CX3CL1-CX3CR1 signaling in NAFLD development remains controversial. In the mouse liver, CX3CL1 is expressed in Kupffer cells/liver macrophages and HSCs [71], while CX3CR1 is mainly expressed in Kupffer cells [71,72]. Sutti et al. reported that CX3CR1-positive monocyte-derived dendritic cells (moDCs) contribute to hepatocyte injury by producing TNF-α in a murine model of steatohepatitis induced by a methionine/choline-deficient (MCD) diet [73] or carbon tetrachloride (CCl4) [74]. Following CCl4 exposure, whole-body CX3CR1-deficient mice (CX3CR1*^gfp^*^/*gfp*^) showed less moDC recruitment into the liver associated with incomplete maturation of monocytes into moDCs [74]. Additionally, in C57BL/6 wild-type mice, treatment with the CX3CR1 antagonist CX3-AT eased CCl4-induced hepatic injury and inflammation along with the decreased moDC accumulation in the liver [75]. These results suggest that CX3CR1 mediates hepatic inflammation by driving moDC recruitment and development. In contrast, Aoyama et al. reported that whole-body CX3CR1 knockout (CX3CR1^−/−^) mice exposed to CCl4 exhibited increased inflammatory cell recruitment into the liver and pro-inflammatory cytokine/chemokine production, including TNF-α, IL-1β, CCL2, and CCL5, but decreased expression of anti-inflammatory IL-10 and arginase-1 in Kupffer cells, resulting in enhanced HSC activation and subsequent liver fibrosis [71]. Additionally, our group reported that CX3CR1^−/−^ mice were more prone to HFD-induced obesity, insulin resistance, and hepatic steatosis and inflammation compared with wild-type control mice [76]. We also found that CX3CL1 expression was lower in the epididymal white adipose tissue (eWAT) of HFD-induced obese C57BL/6J mice, and the long-term (4 weeks) in vivo expression of CX3CL1 by pLIVE^®^ vector (plasma CX3CL1 concentration; 220–250 ng/mL vs. 150–170 ng/mL by empty vector) alleviated insulin resistance and inflammation in the liver and eWAT of obese mice [76]. Collectively, this discrepancy in previous studies may be due to the different roles of the CX3CL1-CX3CR1 signaling in different cell types/tissues.

## 3. Chemokine-Chemokine Receptor Axis as a Therapeutic Target of NAFLD (Small Molecules and Food Factors)

### 3.1. Cenicriviroc (CVC)

Since CCR2 and CCR5 play an important role in the infiltration of myeloid cells and the activation of HSCs, CVC, a once-daily, orally available CCR2/CCR5 dual antagonist, has been expected to improve NASH by suppressing both inflammation and fibrosis, as shown in animal models of steatohepatitis [77,78]. In the Phase 2b CENTAUR study (NCT02217475) of adults with NASH and liver fibrosis (NAFLD activity score ≥ 4 and NASH Clinical Research Network stage 1–3 fibrosis), CVC treatment showed a favorable safety and tolerability profile and improved liver fibrosis without worsening steatohepatitis compared with the placebo [22]. However, and unfortunately, the Phase 3 AURORA study (NCT03028740), which enrolled 1778 participants, of whom 1293 participated in Part 1 of the study [21], was terminated early due to lack of efficacy based on the results of the planned interim analysis of the Part 1 data. The disappointing results of the Phase III trial likely reflect the complexity of the pathogenesis of NAFLD, which involves diverse immune and metabolic pathways. Currently, for CVC, a Phase IIb study has been planned testing the combination therapy with a farnesoid X receptor agonist candidate involving bile acid, cholesterol, and lipid and glucose metabolism [79,80], for treating NASH.

### 3.2. Dietary Carotenoids and Sulforaphane

Many epidemiological studies have demonstrated that the development of NAFLD is closely linked to lifestyle factors (e.g., nutrition, physical activity) [81]. A nutritional intervention with fruits and vegetables could be effective in preventing NAFLD since dietary factors, including antioxidant carotenoids, are useful for decreasing the risk of inflammation-related diseases, including cancer, cardiovascular diseases, and obesity [82,83,84]. β-Cryptoxanthin and lycopene, carotenoids that specifically exist in *Citrus unshiu* (Satsuma mandarin orange) and *Solanum lycopersicum* (tomato), respectively, are relatively abundant in human blood [85,86,87] and have been reported to provide beneficial effects in a murine model of NAFLD. The supplementation of these carotenoids attenuated hepatic lipid accumulation and fibrosis in CL diet- or HFD-fed mice along with the decreased accumulation of T cells in the liver and enhanced anti-inflammatory M2-dominant liver macrophages [88,89,90]. The mechanism of this action was mediated, at least partly, through the downregulation of chemokines, including CCL2, CCL3, CCL5, and CXCL10 [88,89].

Sulforaphane, an isothiocyanate derived from cruciferous vegetables, such as broccoli, is a potent inducer of nuclear factor (erythroid-derived 2)–like 2 (Nrf2), a master transcription factor that regulates oxidative stress responses [90,91]. In addition to its antioxidative effects, sulforaphane has anti-inflammatory properties, suppressing pro-inflammatory IL-8 and CCL2 synthesis by inhibiting NF-κB, STAT6, and MAP kinase pathways [92,93]. We also reported that broccoli extract supplementation mitigated HFD-induced insulin resistance, hepatic steatosis, and the upregulation of CCL2-CCR2 axis [94]. Improved NAFLD by the broccoli extract supplementation was associated with decreased hepatic macrophage accumulation and the M2-dominant polarization of hepatic and adipose macrophages [94]. Additionally, the randomized, placebo-controlled, double-blind trial conducted by Kikuchi et al. demonstrated that supplementation with a dietary dose of broccoli extract for 2 months significantly decreased plasma liver enzymes, ALT, and AST in male participants, suggesting improved fatty liver by sulforaphane [95]. Further nutritional intervention studies, including large, long-term randomized clinical trials with histological assessment of NAFLD, are warranted.

## 4. Conclusions

Accumulating evidence from in vitro and in vivo studies reveals that some chemokine-chemokine receptor axes play a central role in liver inflammation during the development of NAFLD. However, as the results from the AURORA study demonstrate, therapeutic applications for targeting chemokines and chemokine receptors to resolve steatohepatitis and fibrosis are still challenging. Further basic and clinical research is essential for better understanding the molecular mechanisms by which the chemokine system mediates hepatic and adipose inflammation as well as their interaction in the progression of NAFLD. To improve the response rates among patients with NAFLD, combination approaches including lifestyle interventions that are personally tailored to the patient’s disease drivers, such as obesity and type 2 diabetes, are required.

## Figures and Tables

**Figure 1 medicina-58-00761-f001:**
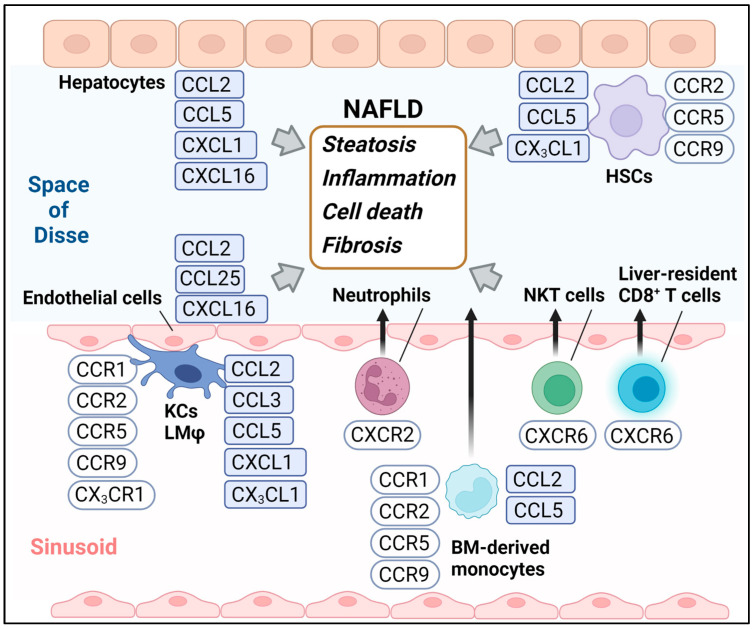
The chemokines and chemokine receptors highlighted in this review. BM, Bone marrow; KCs, Kupffer cells; LMφ, Liver macrophages; HSCs, Hepatic stellate cells.

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
