# Peer review of "An Update on the Chemokine System in the Development of NAFLD"

_medicina, 2022, doi:10.3390/medicina58060761_

Round 1

Reviewer 1 Report

A Systematic Review and Network Meta-Analysis on the relationship between chemokines and NAFLD has been recently published (not cited by the Authors).

Pan X, Chiwanda Kaminga A, Liu A, Wen SW, Chen J, Luo J. Chemokines in Non-alcoholic Fatty Liver Disease: A Systematic Review and Network Meta-Analysis. Front Immunol. 2020 Sep 18;11:1802. doi: 10.3389/fimmu.2020.01802. PMID: 33042108; PMCID: PMC7530185.

This review paper is an update on chemokine system in the development of NAFLD. The paper is well written and requires very few corrections. In particular, it appears that the paper is a summary of their recent research already published. In fact, there are many self citations and this should somehow be corrected or better highlighted.

It is more a question of style than substance. This reviewer compliments the authors on their research.

Author Response

Reviewer#1

Comment#1: A Systematic Review and Network Meta-Analysis on the relationship between chemokines and NAFLD has been recently published (not cited by the Authors). Pan X, Chiwanda Kaminga A, Liu A, Wen SW, Chen J, Luo J. Chemokines in Non-alcoholic Fatty Liver Disease: A Systematic Review and Network Meta-Analysis. Front Immunol. 2020 Sep 18;11:1802. doi: 10.3389/fimmu.2020.01802. PMID: 33042108; PMCID: PMC7530185.

This review paper is an update on chemokine system in the development of NAFLD. The paper is well written and requires very few corrections. In particular, it appears that the paper is a summary of their recent research already published. In fact, there are many self-citations and this should somehow be corrected or better highlighted. It is more a question of style than substance. This reviewer compliments the authors on their research.

Response: As the Reviewer suggested, we have cited the article by Pan et al. in the revised manuscript (Ref. 32). The study supports our findings in the circulating CCL3 levels of patients with NASH. To address the self-citation issue raised by the Reviewer#1, we carefully choose the references again, and removed our two papers from the manuscript. In the Reference of the revised manuscript, 6 papers out of 95 are reported by our group.

Reviewer 2 Report

The authors give a comprehensive update on the current knowledge on the role of the chemokines and their ligands in nonalcoholic fatty liver disease. The paper is well written. I have only a few semantic remarks.

Title: update on the chemokine system

l. 42: are a group

l. 160: were elevated

l. 270: are required (instead of 'must be')

Author Response

Reviewer#2

The authors give a comprehensive update on the current knowledge on the role of the chemokines and their ligands in nonalcoholic fatty liver disease. The paper is well written. I have only a few semantic remarks.

Title: update on the chemokine system

  1. 42: are a group
  2. 160: were elevated
  3. 270: are required (instead of 'must be')

Response: Thank you so much for pointing that out. We have fixed them.